# Soil Enzyme Activity and Microbial Metabolic Function Diversity in Soda Saline–Alkali Rice Paddy Fields of Northeast China

**Yunke Qu** [1,2] **, Jie Tang** [2,*] **, Zhaoyang Li** [2] **, Zihao Zhou** [2] **, Jingjing Wang** [2] **, Sining Wang** [2] **and Yidan Cao** [2]

1   Key Lab of Groundwater Resources and Environment, Ministry of Education, Jilin University, Changchun 130012, China; quyunke1024@163.com

2   College of New Energy and Environment, Jilin University, Changchun 130012, China; zhaoyang@jlu.edu.cn (Z.L.); zhouzh18@mails.jlu.edu.cn (Z.Z.); wangjj16@mails.jlu.edu.cn (J.W.); wangsn16@mails.jlu.edu.cn (S.W.); 18626725081@163.com (Y.C.)

*   Correspondence: tangjie@jlu.edu.cn; Tel./Fax: +86-0431-85159440

**Abstract:** Western Jilin province has the most serious area of soda salinization in Northeast China, which affects and restricts the sustainable development of agriculture. The effects of physico-chemical properties of rhizosphere and non-rhizosphere soil on soil microbial diversity and enzyme activities (polyphenol oxidase, catalase, invertase, amylase) were evaluated in typical soda saline-alkali paddy field. Community-level physiological profile (CLPP) based on Biolog-ECO plates was used to assess the functional diversity of soil microorganisms. Exchangeable sodium percentage (ESP) and pH were negative correlated with the microbial activity (AWCD), soil enzyme activities (amylase, sucrose, and catalase, except for polyphenol oxidase) in rice rhizosphere and non-rhizosphere soil ($P < 0.05$). The indexes of microbial diversity in rice rhizosphere soil were significantly higher than that of non-rhizosphere soil. The utilization of amino acids by rice rhizosphere microorganisms was relatively high, while non-rhizosphere soil had relatively high utilization of carboxylic acid, phenolic acid, and amine. Among the selected physico-chemical properties, soil organic carbon (SOC) and soil water content (SWC) had the greatest influence on the variation of microbial diversity indexes and enzyme activities in rhizosphere soil. ESP and pH showed a significant positive correlation with carbon source utilization, especially for amine (AM) and phenolic acid (PA) carbon source utilization ($P < 0.05$) by means of RDA, and the utilization rate of AM and PA carbon sources by rice rhizosphere and non-root soil microorganisms was P1 < P2 < P3.

**Keywords:** physic-chemical properties; enzyme activities; microbial diversity; utilization of carbon source; saline-alkali paddy soil

## 1. Introduction

Songnen Plain, an important grain-producing area in China, is one of the three major concentrated distribution areas of soda saline-alkali soil in the world [1]. It lies in the central part of Northeast China, which is a typical vulnerable area for global carbon cycle research [2,3]. In recent years, with the acceleration of urbanization and industrialization in China, the population growth demands food and means of livelihood, and the excessive exploitation of natural resources leads to an increasingly arid climate, desertification, salinization, and grassland degradation, which brings great pressure to the limited soil resources [4–6]. Soil salinization will harm the normal growth of plants, changes the structure and function of the cell membrane, and produces toxic effects on cells. At the same time, it increases the osmotic pressure of the soil and causes resistance to the absorption abilities of

the plant, which makes photosynthesis and metabolism of the plant unable to function, resulting in cell dehydration, plant wilting, and finally leading to plant death [7–10]. Soil salinization reduces agricultural productivity and threatens the sustainable development of ecology and the economy. Therefore, effective measures should be taken to reduce soil salinization [11].

　　Lorenz Hiltner, a German scientist recognized as the first scientist to coin the term "rhizosphere" in 1904, defined rhizosphere as a narrow zone of soil, which is influenced by living roots. The species and quantity of rhizosphere microorganisms directly affect the biochemical activity of soil and the composition and transformation of soil nutrients [12,13]. The rhizosphere effect of rhizosphere soil can be expressed by the rhizosphere difference between rhizosphere soil and non-rhizosphere soil under specific environmental and ecological conditions. The most important effect of the rhizosphere effect on rhizosphere microorganisms is nutrient selection and enrichment [14,15]. The exchange of materials and energy among microorganisms, soil, and plants forms a close and special relationship among them, which makes the abundance and species of rhizosphere microorganisms different from that of non-rhizosphere soil to some extent. It is of great significance to study the microbial community structure of plant rhizosphere in saline-alkali land to make full use of saline-alkali land resources and improve land use utilization rate.

　　Soil enzymes are the most active part of soil organic components, which participate in all biochemical processes of soil environment, and are often used as indicators to predict soil ecosystem and environmental quality [16–18]. Previous studies have shown that oxidase activity is more influenced by soil pH, hydrolase activity is more related to soil organic matter content, and directly involved in the mineralization of organic matter, thus affect the nutrient and carbon cycle [19]. In general, the enzyme activity decreased when the soil was wet, but higher when the soil moisture was moderate. When the soil moisture content was low, the activities of protease and cellulase were significantly decreased, while the activities of polyphenol oxidase and peroxidase were decreased with the increase of water content, but the activity of hydrolase was not significantly decreased [20,21]. Zhang et al. found that planting varieties with different salt tolerance had different effects on rhizosphere soil pH, soil available nutrient content, and soil enzyme activities [22]. Wan et al. pointed out that soil pH had a direct impact on soil biochemical reactions participated in soil enzymes [23]. Freeman et al. considered that after the anaerobic environment of the wetland was destroyed, the activities of phenoloxidase and hydrolase increased sharply due to the increase of oxygen content in the soil, which promoted the degradation of soil organic carbon [24].

　　Soil microbial community plays a key role in regulating the material cycle of the ecosystem. However, plant community structure, soil pH, water, organic carbon, temperature, climate, and environmental factors may influence soil microbial community composition and biological activity [25]. At present, the research of plant rhizosphere has won worldwide attention, and the relationship between soil enzyme, soil microorganism, and crop has become one of the key fields in many interdisciplinary studies [26,27]. It has been reported that soil biological activity is highly correlated with soil physical and chemical properties [28,29]. Previous studies mainly focused on the microbial biomass and community structure of plant rhizosphere soil and found that different rhizosphere soil microbial characteristics were obtained due to different regions, environments, land use patterns, and growth time [30]. However, little is known about the effects of soil microbial community metabolic functional diversity in saline-alkali soil areas. The analysis of rhizosphere microecological characteristics, including soil enzyme activity and microbial diversity, can provide a theoretical basis for crop cultivation from the perspective of microecology.

　　We hypothesized that saline-alkali soil affects the abundance and diversity of soil bacteria and the utilization capacity of C by influencing the physical-chemical properties of soils. The carbon source metabolic capacity in Biolog Eco-plates of the microbial community could be assessed by its community-level physiological profile (CLPP) [31]. The aim of this study is to detect the effects of environmental factors on the microbial activity and metabolic function diversity of rhizosphere and non-rhizosphere soil enzyme activities (amylase, sucrase, catalase, and polyphenol oxidase) in different

saline paddy fields. It is of great significance for regional soil resource protection, environmental management, and agricultural sustainable development to understand the main indicators affecting the rhizosphere soil microorganism and soil quality in the agricultural ecosystem of the saline-alkali fields in the west of Jilin Province.

## 2. Materials and Methods

### 2.1. Study Area Description

Western Jilin province is located at 123°09′–124°22′ E, 44°57′–45°46′ N, where it has experienced multiple desertification and saline-alkali desertification of evolution process [32]. It belongs to the global change research transect in Northeast China's typical semi-arid and semi-humid continental monsoon climate [33]. The annual average rainfall is approximately 400 mm and is concentrated in July and August [34]. Affected by the winter wind, it is cold in winter, with more snowfall but less evaporation. The freezing period generally lasts from late October to April of the next year, during which the depth of the frozen soil layer usually reaches 1.2 m to 1.5 m [33].

### 2.2. Soil Sampling and Experimental Design

The study site was located in Songyuan, Jilin province, China (Figure 1). According to a soil type map and utilization map, three paddy fields (P1, P2, P3) with different saline-alkali stresses were selected after pre-experiments and initial data analysis of the project (Table 1). Representative plots were distributed in the main irrigation regions in Western Jilin Province and the soil types were all saline-alkali meadow soils. The plots were covered with native degraded grass vegetation (Leymus chinensis, C3 plant) before crop cultivation, without any prior cultivation or fertilization. All the fields were fertilized once on May 8th and managed in the same way.

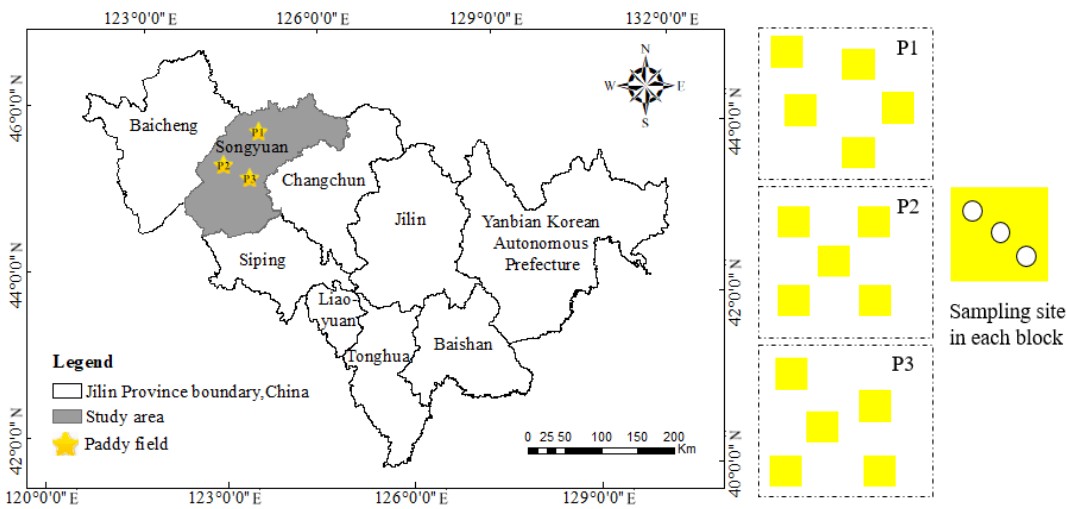

**Figure 1.** The location of the study area (Songyuan) in Jilin province, in Northeast China.

**Table 1.** Background information on sampling sites.

| Sampling Site | pH | ESP (%) | EC (ds cm⁻¹) | Soil Classification | Clay (%) | Silt (%) | Sand (%) | Fertilization Treatment |
|---|---|---|---|---|---|---|---|---|
| P1 | 8.23 | 5.85 | 0.35 | Loam | 14.07 | 40.18 | 45.75 | N, P and K |
| P2 | 9.37 | 8.09 | 0.31 | Silty loam | 10.75 | 67.33 | 21.92 | N, P and K |
| P3 | 9.94 | 13.20 | 0.45 | Sandy loam | 4.58 | 31.30 | 64.12 | N, P and K |

The rhizosphere (R) and non-rhizosphere soil samples (N) were collected on 16 August 2018 (vigorously growing stage). We established five blocks (15 m × 15 m) in each field randomly, and selected

three parallel sampling site within each block (Figure 1). We inserted the iron sheet with no bottom into the sampling point and removed the water inside, the plants were dug out with a spade (attention paid to the integrity of the root system). The loose soil shaken off from the roots was used as non-rhizosphere soil, and then the thin layer of soil attached to the root of rice was scraped with flame sterilized tweezers. Due to the difference between the paddy field and upland, we assumed that the soil material attached to the roots was considered as the rhizosphere soil of rice. The spade was cleaned with distilled water before each sampling to avoid contamination among samples. Subsamples from the three sampling sites were homogeneously mixed to generate a composite sample and immediately put into a sterile sealed bag, stored in an incubator filled with ice, quickly transported to the laboratory, stored at 4 °C in the refrigerator before being analyzed and tested.

### 2.3. Soil Analysis

Soil physico-chemical properties were measured following the methods by Zheng [35].

Soil water content (SWC) was determined on fresh soils by measuring weight loss after drying at 105 °C for 24 h. The sampled soils were dried in an air-circulating room, and then stones and plant residues were removed to obtain homogenous samples. Half of the sampled soils passed through a 2 mm nylon sieve for measurement of soil properties, the other half were ground again and sieved for soil organic carbon (SOC) assay. Soil pH and electrical conductivity (EC) were determined by pH meter (soil: distilled water = 1:5)

Exchangeable sodium percentage (ESP) was calculated as:

$$ESP = Na^+/CEC \times 100\%$$

where $Na^+$ is the concentration of exchangeable sodium (c mol $(Na^+)$ kg$^{-1}$) and CEC is cation exchange capacity (c mol kg$^{-1}$). Exchangeable $Na^+$ concentration was assayed by using flame photometry (Shimadzu optical doublebeam atomic absorption spectrophotometer, Shanghai). CEC was measured through the EDTA-ammonium acetate salt exchange method.

The SOC content was determined by using a total organic carbon analyzer (Shimadzu TOC-V, Kyoto, Japan) with the SSM-5000A module. The content of total carbon and inorganic carbon was calculated separately and the difference was reported as the SOC (%).

### 2.4. Soil Enzyme Activities

Soil enzyme activities were determined using spectrophotometry [36]. Amylase (AMY, EC 3.2.1.2) activity was determined by using phosphate buffer (pH = 5.5, 10 mL) and substrate (10 mL 1% starch solution). After incubation at 37 °C for 24 h, the content of maltose was measured at 508 nm.

Invertase (INV, EC 3.2.1.26) was determined by using phosphate buffer (pH = 5.5, 15 mL) and substrate (5 mL 8% sucrose solution for invertase). The amount of glucose was measured at 508 nm after incubation at 37 °C for 24 h.

Catalase (CAT, EC 1.11.1.6) activity was determined by measuring the consumption of $H_2O_2$ via potassium permanganate titration. Added to 5 mL 0.3% $H_2O_2$ and 40 mL distilled water was 2 g dried soil, which was vibrated for 20 min. Then, 5 mL 3 N $H_2SO_4$ was added at the end and the reaction mixture needed to be filtered immediately. After that, 25 mL filtrate was taken to titration with 0.1 N $KMnO_4$ to determine the amount of $H_2O_2$.

Polyphenol oxidase (PPO, EC 1.10.3.1) was also determined by the colorimetric method [37]. 1 g soil and 10 mL of 1% pyrogallol solution were added into a 50 mL Erlenmeyer flask and incubated at 30 °C for 2 h, and then citric acid phosphate buffer solution (pH = 4.54 mL) was added. Finally, the solution was extracted through ethyl ether and was measured at 430 nm. According to the standard curves of potassium dichromate standard solution (mg g$^{-1}$), the amount of released purpurogallin in the ethyl ether phase was obtained and used to calculate the enzyme activity.

### 2.5. Biolog—ECO Plate Assay

A community-level physiological profile (CLPP) was obtained to evaluate the metabolic functional diversity of microorganisms based on the different characteristics of substrates that could be utilized by the microorganisms [38]. ECO plate consists of 3 parallel groups, 31 carbon substrates in each group, 3 blank controls, and tetrazolium dye as substrate utilization indicator [39]. Equivalents of 10 g dried soil and 90 mL 0.85% sterilized NaCl solution (pH = 7) were added into the sterilized Erlenmeyer flask and sealed. They were then shaken fully at 200 rpm on Orbital Shaker Incubator at 25 °C for 40 min and allowed to settle for approximately 30 min to decant soil particles. The soil bacterial suspensions were diluted ($10^{-2}$) in sterilized NaCl solution. Then, 150 μL diluted suspensions were added to each of the Biolog-Eco plates well using an 8-channel pipette incubated in the dark at 25 °C for 9 days and the substrate utilization was measured at 590 nm (μQuant spectrometer; BIO-TEK Instruments, Winooski, VT, USA). The first optical density (OD) was measured immediately after inoculation, and then every 24 h for 216 h.

### 2.6. Statistical Analysis

The study used the Microlog4.2 software platform to process Biolog data and convert format for analysis. One-way analysis of variance (ANOVA) was computed to analyze the significant differences of soil physic-chemical properties, enzyme activities, and microbial diversity indicators using the SPSS for Windows version 19.0 (SPSS Inc., Chicago, IL, USA; Norusis, 2008). Significance was evaluated at $P < 0.05$ using Duncan's test. The obtained data were represented by mean ± standard deviation (Mean ± SD), and the distribution of data was represented by error bars. The Origin Graph 8.5 software package was used to draw the Graph. Redundancy analysis (RDA) was carried out by Canoco5 software (Microcomputer Power, Inc., Ithaca, NY, USA).

The metabolic intensity of microorganisms can be reflected by the average well color development (AWCD) and the microbial diversity indexes were calculated using the following formula:

$$\text{AWCD} = \sum (C - R)/31 \tag{1}$$

where $C$ is the OD value of each well and $R$ is the OD value of the Biolog-Eco plates control well.

Soil microbial diversity can be estimated by Shannon index (H), Simpson index (D), and evenness index (E) calculated by the OD values at 96 h using the following formulae [40,41]:

$$\text{Shannon index}: \ \text{H} = - \sum (\text{P}_i \times \ln \text{P}_i); \tag{2}$$

$$\text{Evenness index}: \ \text{E} = \ \text{H}/\text{H}_{max} \ = \text{H}/\ln \text{S}; \tag{3}$$

$$\text{Simpson index}: \ \text{D} = 1 - \sum \text{P}_i{}^2; \tag{4}$$

where $\text{P}_i$ represents the ratio of the relative absorption value of the well to the total relative absorption value (AWCD) of the entire Biolog-Eco plates, S represents the number of carbon sources that can be utilized by microorganisms.

## 3. Results

### 3.1. Soil Physico-Chemical Properties

The rhizosphere (R) and non-rhizosphere (N) soil physico-chemical properties of the paddy farmlands were summarized in Table 2. The mean pH values of rhizosphere and non-rhizosphere soils were P1 (8.22) < P2 (9.05) < P3 (9.45). This was positively correlated with soil ESP values P1 (6.49%) < P2 (7.46) < P3 (13.20%). The content of SOC decreased with the increase of soil pH and ESP. When comparing the rhizosphere (R) and non-rhizosphere soil, it was found that the pH and ESP

of rhizosphere soil were lower than that of non-rhizosphere soil, while values of EC, SWC, and soil organic carbon content were opposite.

**Table 2.** Physico-chemical properties of three paddy soils. The data are mean values of the rhizosphere (R) and non-rhizosphere (N) soil respectively. Significant differences analyses between rhizosphere (R) and non-rhizosphere soil were based on one-way ANOVA followed by the Fisher LSD test. Lowercase letters mean a significant difference in different parts of plants in the same place ($P < 0.05$) and capital letters mean a significant difference in different parts of plants in the same place ($P < 0.05$).

| Sampling Site | P1R | P1N | P2R | P2N | P3R | P3N |
|---|---|---|---|---|---|---|
| pH | 8.07 ± 0.16 Aa | 8.37 ± 0.05 Ab | 8.91 ± 0.12 Ba | 9.19 ± 0.14 Bb | 9.33 ± 0.16 Ca | 9.57 ± 0.15 Cb |
| ESP (%) | 6.35 ± 0.06 Aa | 6.63 ± 0.08 Bb | 7.45 ± 0.11 Ba | 8.07 ± 0.13 Bb | 12.21 ± 0.23 Ca | 14.18 ± 0.31 Cb |
| EC (dS m$^{-1}$) | 0.39 ± 0.07 Aa | 0.28 ± 0.05 Ab | 0.27 ± 0.06 Aa | 0.36 ± 0.07 Bb | 0.53 ± 0.07 Ba | 0.41 ± 0.01 Bb |
| SWC (%) | 0.54 ± 0.06 Aa | 0.38 ± 0.03 Ab | 0.50 ± 0.06 Aa | 0.41 ± 0.09 Ab | 0.51 ± 0.21 Aa | 0.32 ± 0.19 Ab |
| SOC (g kg$^{-1}$) | 12.35 ± 0.11 Aa | 10.31 ± 0.17 Ab | 11.27 ± 0.07 Ba | 8.36 ± 0.16 Bb | 9.58 ± 0.23 Ca | 7.49 ± 0.27 Cb |

## 3.2. Soil Enzyme Activities

The activities of amylase, invertase, and catalase in the rhizosphere were 1.63, 1.92, and 1.62 times higher than those in non-rhizosphere soil, respectively. Alternatively, the activity of polyphenol oxidase in non-rhizosphere soil was 1.82 times higher than that in the rhizosphere in both three saline paddy fields (Figure 2). ANOVA revealed a significant difference ($P < 0.05$) between the rhizosphere and non-rhizosphere soil with higher values of rhizosphere soil than that of non-rhizosphere soil, except for polyphenol oxidase. The correlation analysis of enzyme activities and soil physico-chemical properties was shown in Table 3 ($P < 0.05$).

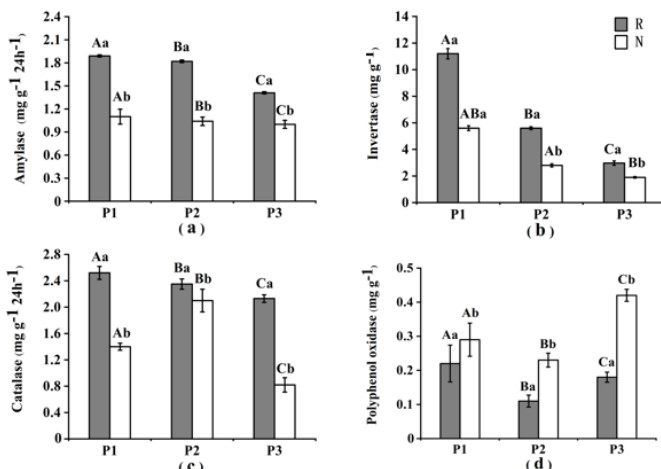

**Figure 2.** Soil enzyme activities in different sites (R = rhizosphere and N = non-rhizosphere): (**a**) Amylase; (**b**) Invertase; (**c**) Catalase; and (**d**) Polyphenol oxidase. Error bars represent the 95% confidence interval ($n = 15$). Capital letters indicate significant differences between different paddy fields, while lowercase letters indicate differences between rhizospheric and non-rhizospheric soils in the same paddy field ($P < 0.05$).

**Table 3.** Pearson correlation coefficient (r) between soil physico-chemical properties and enzymeactivities and microbial diversity indexes in paddy fields.

|  | AMY | INV | CAT | PPO | H | E | D |
|---|---|---|---|---|---|---|---|
| pH | −0.53 | −0.92 * | −0.431 | 0.263 | −0.744 | −0.458 | −0.887 * |
| ESP | −0.45 | −0.71 | −0.59 | 0.51 | −0.62 | −0.34 | −0.76 |
| EC | −0.08 | −0.26 | 0.01 | 0.11 | −0.14 | 0.15 | −0.34 |
| SWC | 0.88 * | 0.62 | 0.93 * | −0.84 * | 0.87 * | 0.93 ** | 0.71 |
| SOC | 0.89 * | 0.90 * | 0.71 | −0.63 | 0.96 ** | 0.83 * | 0.97 ** |

* Correlation is significant at 0.05 level (2-tailed); ** Correlation is significant at 0.01 level (2-tailed).

### 3.3. Community-Level Physiological Profile

AWCD reflects soil microbial activity and higher AWCD indicates higher soil microbial activity [42]. Figure 3 shows that the AWCD of rice rhizosphere and non-rhizosphere soil microorganisms increased gradually with the extension of culture time. During the same incubation period, the AWCD of rhizosphere soil microorganisms was higher than that of non-rhizosphere soil and did not change significantly at the beginning of incubation. However, after 24 h, AWCD increased rapidly and after 196 h of incubation, AWCD gradually stabilized. The overall trend of AWCD in the rhizosphere and non-rhizosphere soils was the same, both of which increased first and then gradually leveled off. With the increase of salinity degree, AWCD values of the rhizosphere and non-rhizosphere of three paddy fields were P1 > P2 > P3 (Figure 3).

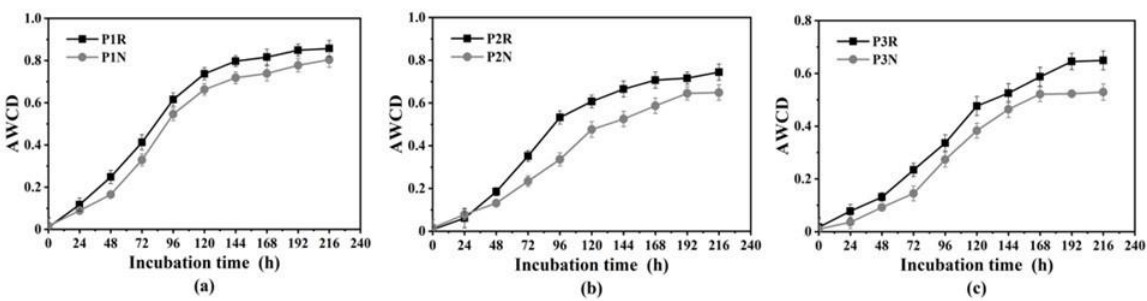

**Figure 3.** The average well color development (AWCD) of rhizosphere and non-rhizosphere soil microorganisms in three paddy fields: (**a**) P1, (**b**) P2, and (**c**) P3.

### 3.4. Functional Diversity Index of Soil Microbial Community

The soil microbial Shannon index (H), Evenness index (E), and Simpson index (D) were calculated according to the carbon source metabolism of culturable microorganisms in 96 h, which can accurately reflect the diversity characteristic function of this fraction of the microbial community. The results were shown in Table 4. Both rhizosphere and non-rhizosphere soil microbial diversity indexes in saline-alkali paddy fields were different. The microbial diversity index of P1, P2 and P3 rhizosphere soil was 2.71, 2.08, 1.45 (P1 > P2 > P3), and the microbial diversity index of non-rhizosphere soil was 1.28, 0.91, 0.61 (P1 > P2 > P3), respectively. When comparing the soil microbial diversity index of different saline-alkali paddy fields, ANOVA revealed a significant difference ($P < 0.05$). Shannon index (H), Evenness index (E), and Simpson index (D) were significantly positively correlated with each other. Except for polyphenol oxidase, pH and ESP were negatively correlated with soil enzyme activities and microbial diversity indexes. It showed that soil SWC and SOC were the most important physico-chemical factors affecting the variation of enzyme activities and microbial diversity indexes in our study sites (Table 3).

**Table 4.** Soil microbial diversity index of rhizosphere and non-rhizosphere of saline-alkali rice (96 h). The data in the table are mean value ± standard deviation, Lowercase letters mean a significant difference in different parts of plants in the same place ($P < 0.05$) and capital letters mean a significant difference in different parts of plants in the same place ($P < 0.05$).

| Field | Shannon-Wiener Index (H) | | Evenness Index (E) | | Simpson Index (D) | |
|---|---|---|---|---|---|---|
| | R | N | R | N | R | N |
| P1 | 2.71 ± 0.38Aa | 1.28 ± 0.11Ab | 0.88 ± 0.02Aa | 0.29 ± 0.02Ab | 0.80 ± 0.08Aa | 0.55 ± 0.05Ab |
| P2 | 2.08 ± 0.56Ba | 0.91 ± 0.12Bb | 0.76 ± 0.01Ba | 0.28 ± 0.02Ab | 0.63 ± 0.15Ba | 0.42 ± 0.36Bb |
| P3 | 1.45 ± 0.04Ca | 0.61 ± 0.27Ab | 0.64 ± 0.02Ca | 0.26 ± 0.03Ab | 0.44 ± 0.02Ca | 0.35 ± 0.28Ca |

### 3.5. Soil Microbial Utilization Intensity of Carbon Source

According to the properties of chemical groups, the carbon substrates could be predominantly classified into six categories, namely, carbohydrates (CH) (10), amino acids (AA) (6), carboxylic acids (CA) (7), polymer (PM) (4), amine (AM) (2), and phenolic acids (PA) (2) (Table 5) [43,44]. The relative utilization of carbon sources was higher in CH, AA, and PM, and lower in PA and CA (Figure 4). The utilization of AA in rhizosphere soil was higher than that in non-rhizosphere soil microorganisms, while the utilization of CA, PA, and AM carbon sources in non-rhizosphere soil was higher than that in rhizosphere soil. The utilization of AA and CA in rhizosphere soil increased with the increase of alkalinity, but the results were reversed in non-rhizosphere soil. The utilization of AM and PA in both rhizosphere and non-rhizosphere soils increased with the increase of alkalinity. It can be seen that salinity stress had different effects on the utilization of carbon source types by soil microorganisms.

**Table 5.** Types and distribution of the carbon sources in Biolog-Eco plate (96 wells).

| Horizon | 1 [a]–5 [a]–9 [a] | 2 [a]–6 [a]–10 [a] | 3 [a]–7 [a]–11 [a] | 4 [a]–8 [a]–12 [a] |
|---|---|---|---|---|
| A [b] | Water | CH:β-methyl-D-glucoside | CH:D-galactonic acid γ-lactone | AA:L-arginine |
| B [b] | CA:Pyruvatic acid methyl ester | CH:D-xylose | CA:D-galacturonic acid | AA:L-asparagine |
| C [b] | PM:Tween 40 | CH:I-erythritol | PA:2-hydroxybenzoic acid | AA:L-phenylalanine |
| D [b] | PM:Tween80 | CH:D-mannitol | PA:4-hydroxybenzoic acid | AA:L-serine |
| E [b] | PM:α-cyclodextrin | CH:N-acetyl-D-glucosamine | CA:R-hydroxybutyric acid | AA:L-threonine |
| F [b] | PM:Glycogen | CA:D-glucosaminic acid | CA:D-galacturonic acid | AA:Glycyl-L-glutamic acid |
| G [b] | CH:D-cellobiose | CH:Glucose-1-phosphate | CA:a-ketobutyric acid | AM:Phenyl ethylamine |
| H [b] | CH:α-D-lactose | CH:D,L-α-glycerol phosphate | CA:D-malic acid | AM:Putrescine |

Note: [a] Column numbers of the 96-well plate; [b] Row numbers of the 96-well plate; CH: carbohydrate; AA: amino acids; CA: carboxylic acids; PM: polymer; PA: phenolic acid; AM: amine. The same below.

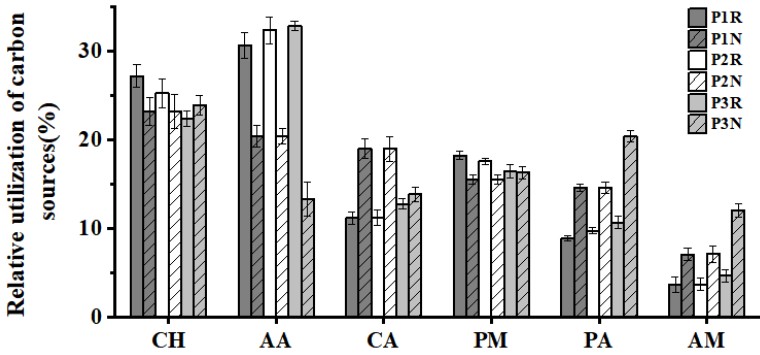

**Figure 4.** Utilization intensity of 6 types of carbon sources in the rhizosphere and non-rhizosphere soil microbial communities of saline-alkali rice. CH: carbohydrate; AA: amino acids; CA: carboxylic acids; PM: polymer; PA: phenolic acid; AM: amine. The same below.

The carbon source utilization rate of rhizosphere and non-rhizosphere soils was controlled by soil physicochemical parameters. Soil pH, ESP, SOC, SWC, EC were the major determinants of the relative utilization of carbon sources. The RDA analysis showed that environmental factors accounted for the utilization of different carbon sources of soil microorganisms on the axes (RDA1 97.07%, RDA2 2.85%) (Figure 5). All data were the mean values of samples from 3 composite samples in 5 blocks of each sampling site ($n = 15$). The carbon source utilization of microorganisms was generally separated by rhizosphere soil and non-rhizosphere soil. The non-rhizosphere sample points were separated along the second coordinate axis. According to the length of the arrow, SWC and SOC have a great influence on the utilization of microbial carbon sources in rhizosphere soils. ESP, pH, and the arrow of 6 types of carbon sources show an acute angle, indicating a positive correlation between them, and had an especially significant impact on AM and PA carbon sources. The distance of sample points in the same group indicated the strength of sample repeatability, while the distance of samples in different groups reflected the difference of sample distance between groups. The sample distance with strong heterogeneity was farther apart.

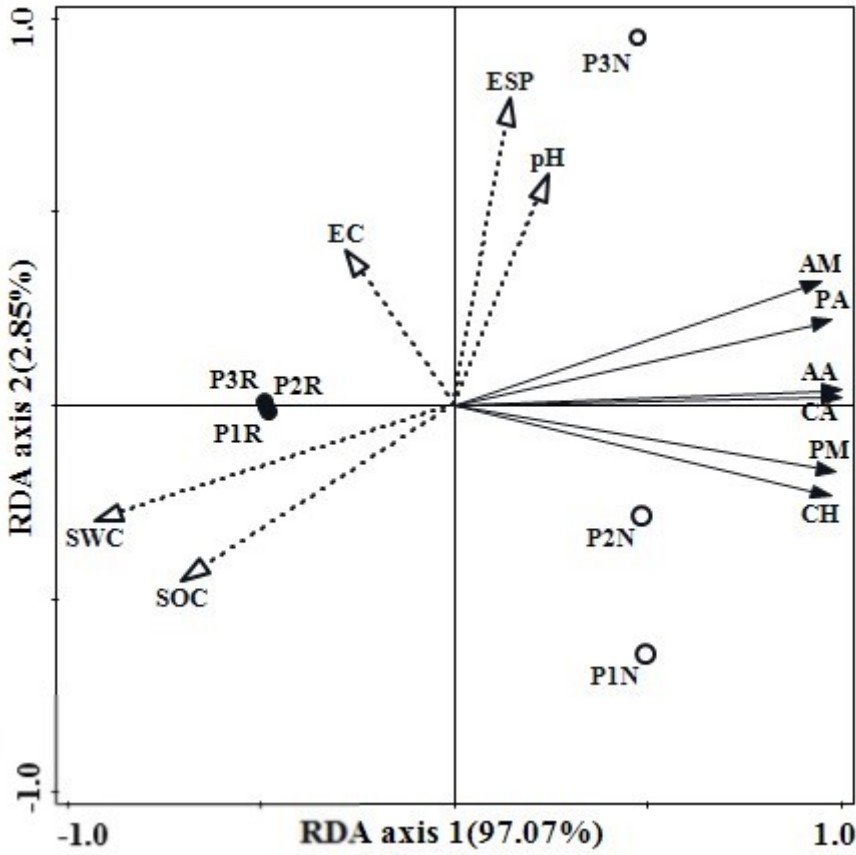

**Figure 5.** Redundancy discrimination analysis (RDA) depicting the relationship between the main soil physicochemical parameters and carbon source utilization rate of rhizosphere and non-rhizosphere soils. The categories with unique symbols demonstrated a significant position in the multivariate space (RDA; 499 Monte Carlo permutations. *P* < 0.05). Black solid circle (•) rhizosphere soil sample; Black empty circle (○) non-rhizosphere soil sample. Use the solid arrow for the relative utilization of substrate carbon sources; the dotted arrows for the major environmental factors.

## 4. Discussion

The formation of soil is influenced by soil parent material, topography, climate, organisms, and human activities [45]. Plants provide litter and root exudates for the soil decomposer community as sources of carbon and nutrients, and regulate the chemical properties of soil solutions [46]. Soil moisture, structure, fertility, pH, and salinity are the main factors affecting the distribution and

species composition of rhizosphere and non-rhizosphere soil communities [47]. The physico-chemical properties of rhizosphere and non-rhizosphere soil in different saline-alkali rice fields were obviously different (Table 2). Most farmlands in western Jilin province are saline-alkali lands with the concentration of $Na_2CO_3$ and $NaHCO_3$ is relatively high. Even when their content is very low (0.5 c mol·$L^{-1}$), it can cause soil salinization and the pH value to increase significantly, which will directly damage the plant root tissue and cause the loss of normal physiological function of root cells. At the same time, high pH reduces the availability of phosphorus, weakens the uptake of nutrients by rice roots, and affects the normal growth and development of plants [48]. Soda saline-alkali soil has the characteristics of high bulk density, low porosity, and preventing water infiltration when absorbing water. When the ESP is too high, the sodium colloid will swell and shrink when losing water. Excessive exchangeable sodium will replace other cations and trace elements absorbed by the soil, such as $Ca^{2+}$ and $Mg^{2+}$ [49,50], resulting in decreased fertility and quality of soil. In addition, we also found that the effect of soil pH on organic carbon is complex, and soil microbial biomass and community structure, as well as enzyme production and secretion, are affected by pH [51,52]. In general, the optimal pH condition of most bacteria and actinomycetes is 6.5–8, and that of fungi is 5–6. In addition, in an environment where pH is too high (or too low), the microbial activity will be reduced, and the enzymes which play an important role in the transformation of nutrients and humus formation will be inactivated, which may slow down the humification of the litter, resulting in the loss of soil fertility and organic carbon content [53]. Measuring the concentration of enzymes found in the rhizosphere and non-rhizosphere soil including hydrolase (amylase), invertase, and oxidordeuctase (catalase and polyphenol oxidase) gives insight into the biochemical reaction of soil [49]. Catalase is most widely distributed in soil and can break down hydrogen peroxide to prevent its toxic effects on living organisms [54]. Invertase catalyzes sucrose hydrolysis and plays an important role in increasing the accessibility of nutrients in the soil. Amylase is mainly derived from microorganisms, which hydrolyze starch to produce reducing sugars and is an important energy source for microorganisms [55]. Our findings that values of amylase, invertase, and catalase in rhizosphere soil were higher than that in non-rhizosphere soil (Figure 1). The high enzyme activity in rhizosphere soil may be related to the larger microbial population, which is helpful for the release of these enzymes [56]. Plant roots may alter microbial communities by altering water fluxes and oxygenation to increase soil porosity and provide sufficient nutrients to the soil by releasing exudates [57]. PPO can promote the degradation of refractory phenolic compounds and is one of the most important factors in SOC decomposition [51]. The activity of polyphenol oxidase in non-rhizosphere is higher than that in rhizosphere soil, which may be related to the involvement of polyphenol oxidase in decomposition and synthesis during humification, and it is a medium of humus [58]. Polyphenol oxidase is involved in the transformation of aromatic compounds in soil organic humus, and plays an important role in the transformation of soil aromatic organic compounds into humus, which is mainly secreted and synthesized by microorganisms. Compared with the other three enzymes, the main reason for the low content of polyphenol oxidase may be that the hydroxyl ion can react with the active center of the enzyme under alkaline conditions, resulting in the loss of enzyme activity [59]. Hao et al. (2006) found that the activity of polyphenol oxidase was 1.6 times of that in hypoxia [60]. Therefore, it may also be due to the hypoxia of soil environment under long-term flooded paddy fields, which leads to the low activity of polyphenol oxidase compared with other enzymes.

Average well color development (AWCD) is an important index to reflect the metabolic activity of soil microorganisms. AWCD of rhizosphere and non-rhizosphere soil was not obvious in the first 24 h, almost less than 0.1, indicating that the carbon source was basically not used by the microbial community. During 48–120 h, it showed a rapid growth trend, indicating that the stronger the utilization of carbon source, the higher the microbial metabolic activity (Figure 3). Shannon index (H) is a comprehensive index reflecting the number and distribution of species and individuals, and it is one of the most widely used indicators of community diversity. Evenness index (E) reflects the average activity level of microorganisms using all carbon source substrates; Simpson index (D) reflects the dominance of common species. The Shannon index, evenness index (E), and Simpson index

of microorganisms in rhizosphere soil were significantly higher than those in non-rhizosphere soil (Table 4), which was similar to the previous research results [61]. Perez Montan et al. thought that the symbiotic system of rhizobia could complete biological nitrogen fixation well, thus increasing the number of rhizosphere microorganisms. In the limited farmland of Songnen Plain, soil bacteria constantly compete and eliminate according to their own needs for nutrients and other resources, which makes changes in the number, quality, and functionality of the community, and these changes of soil bacteria can also affect plant functional groups and their diversity. The growth and development of plants, litter, and root exudates have a certain impact on soil bacteria [62].

　　The results of our Biolog-Eco microplate showed that the AWCD value of rhizosphere soil microorganisms in rice was relatively high (Figure 3), indicating that the carbon source utilization rate of non-rhizosphere soil microbes was low and the functional diversity was thus lower [42,63]. To better understand the metabolic characteristics of a microbial community, it is necessary to analyze not only the utilization mode of microorganisms of the carbon source but also the absolute utilization of a carbon source and its ecological significance. In our study, the utilization rate of amino acids in rhizosphere soil microorganisms was higher, which may be due to the fact that amino acids contain many functional groups, which become highly active bioactive substances after being activated, which have a special role in promoting the development of plant roots. However, the utilization of carboxylic acid, phenolic acid, and amine carbon sources by non-rhizosphere soil microorganisms was relatively high. Carboxylic acid carbon sources account for only a small part of soil soluble organic carbon, but they are important energy sources for soil microbial growth and metabolism [64]. The degradation of complex compounds such as carboxylic acids, phenolic acids, and amines requires the joint action of multiple extracellular enzymes. However, more carbon sources and energy should be put into the synthesis of extracellular enzymes in the growth process of rhizosphere soil microbes, thus reducing the utilization rate of complex compounds by microorganisms [65]. Plant roots and plant residues provide a suitable place and material source for rhizosphere soil microorganisms to grow. The more carbohydrate that plants secrete into rhizosphere soil, the stronger ability of rhizosphere microorganisms to utilize carbon substrates. Rhizosphere soil has been in the environment of exogenous carbon source input (root exudates) for a long time, and it has a certain adaptability to the input of exogenous carbon source. However, the input of exogenous carbon source in non-rhizosphere soil will cause the positive excitation effect of original organic carbon decomposition, and then the mineralization of organic carbon by microorganisms will be enhanced [66]. Plant roots and plant residues provide a suitable place and material source for rhizosphere soil microorganisms to grow. The more carbohydrates plants secrete to rhizosphere soil, the stronger the ability of rhizosphere microorganisms to utilize carbon substrates. The utilization of amino acids and carboxylic acids in rhizosphere soil increased with the increase of alkalinity, but the results were reversed in non-rhizosphere soil (Figure 3) which indicated that saline-alkali soil has a significant regulatory effect on plants, which can change the availability of water and nutrients by plants and microorganisms [67]. With the increase of soil salinity, the growth of some microorganisms sensitive to the change of environment was inhibited, soil microbial community structure and functional diversity were changed, resulting in the different metabolic intensity of carbon sources [68]. Soil salinization is the main factor to inhibit microbial activity and metabolic intensity, which agreed with Chen and Zhao et al. [36,69].

　　It can only reflect the activity of fast-growing or eutrophic microorganisms in the soil, but not slow-growing or unculturable microorganisms [70]. Therefore, this method can only analyze the characteristics of the soil microbial community, and other research methods are needed to comprehensively analyze the structure of the soil microbial community. The later research can further explore the role and mechanism of rice root exudates in regulating microbial community structure, functional diversity, and the role of specific carbon sources in the exudates in microbial-soil. In addition, the Biolog-ECO microplate contains at least nine carbon sources similar to the common rhizospheric exudates, the method can be combined with other technologies such as Liquid chromatography-electrospray ionization mass spectrometry (LC-ESIMS) to analyze the components of

rice rhizospheric exudates and extend them to other plant microbial interactions. In order to further reveal the ecological process of saline-alkali paddy soil, it is necessary to study the microbial diversity of saline-alkali soil by molecular biology.

## 5. Conclusions

In this study, the effects of different physical-chemical properties of rhizosphere and non-rhizosphere on soil enzyme activity and utilization of carbon source of saline-alkali paddy fields (P1, P2, P3) in Northeast China were evaluated by field sampling and laboratory experiments, the following conclusions were drawn:

(1) Saline-alkali soil affected soil microbial diversity by affecting the physical-chemical properties of rice rhizosphere and non-rhizosphere soil. There are differences in soil enzyme activity and soil microbial activity between rice rhizosphere and non-rhizosphere. The enzyme activities and microbial activities of rice rhizosphere and non-rhizosphere soil are different in different degrees. AWCD and microbial diversity indexes of rice rhizosphere soil were higher than that of non-rhizosphere soil.

(2) Saline-alkali soil affected the ability of microorganisms to use carbon sources by affecting the physical-chemical properties of rice rhizosphere and non-rhizosphere soil. Redundancy analysis (RDA) showed that the environmental factors SWC and SOC were the main factors affecting the correlation between community distribution and species distribution in rhizosphere soil in saline and alkaline rice fields. There was a positive correlation between ESP, pH, and carbon sources, especially for AM and PA carbon sources. The rhizosphere soil microorganisms in saline-alkali rice paddy had a higher utilization rate of AA, while the non-rhizosphere soil microorganisms had a higher utilization rate of CA, PA, and AM carbon sources.

**Author Contributions:** Y.Q. and Z.L. drafted the manuscript and revised by J.T., Z.Z. and Y.C. designed the experiments; Y.Q., J.W. collected and tested the samples and S.W. analyzed the data. All authors have read and agreed to the published version of the manuscript.

**Funding:** The program is supported by the Specialized Research Fund for Doctoral Program of Higher Education of China (20130061110065) and the National Natural Science Foundation of China (No. 41471152, 51179073).

**Conflicts of Interest:** The authors declare no conflict of interest.

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
