# Peer review of "Soil Enzyme Activity and Microbial Metabolic Function Diversity in Soda Saline–Alkali Rice Paddy Fields of Northeast China"

_sustainability, doi:10.3390/su122310095_

Round 1
Reviewer 1 Report
Comments to the Author
GENERAL COMMENTS
The manuscript entitled: „The Influence of Rhizosphere and Non-rhizosphere Soil Physico-Chemical Properties on Enzyme Activities and Microorganism in Saline–Alkali Paddy Fields of Northeast China” provided a lot of insightful information on changes in the soil structure of the bacterial community in Saline–Alkali Paddy Fields of Northeast China. In the introduction, the reasons for the salinity of the studied area are presented, which is valuable information. The authors used a correct research methods and received interesting results which were correctly interpreted and statistically developed. The research methodology is described in great detail, which is the value of this manuscript. Unfortunately, in my opinion, changes are required for manuscript to be published. I think the introduction and discussion are the weakest parts of the manuscript.
SPECIFIC COMMENTS
Chapter „Introduction”
In my opinion, in the introduction, too little attention was paid to even a brief description of the enzymes tested, which appears in the discussion, but in my opinion there is space for it here. Moreover, the researchers did not present the reader with the scale of the problem, the harmful effects on the activity of the microbiome. What happens to the bacterial cell, what biochemical processes occur during strong salinity? It is necessary to delve deeper into the functioning of microorganisms in such difficult conditions.
Chapter “Discussion”
In turn, this part of the work lacks a precise answer to several questions. Why did phenol oxidase react differently from the other enzymes?
It would be most appropriate to explain its participation in the biodegradation processes of organic compounds. There is no reliable answer here.
The whole manuscript is also lacking, but especially the discussion lacks consideration of specific groups of microorganisms, species of microorganisms, and their characteristics. Why do those listed deal with salinity pressures? What species or other taxonomic levels cannot cope with this factor? The title of the publication concerns the microbiome of the rhizosphere and the soil beyond, and not really insightful information on their activity in the soil. I will emphasize that the research methods were interesting and obliging to deeper scientific research.

Author Response
Thank you very much for your comments, We have provided a point-to-point response to the changes made in the text. Please see the attachment.

Reviewer 2 Report
MDPI. Sustainability
The influence of rhizosphere and non-rhizosphere soil physiochemical properties on enzyme activities and microorganisms in saline-alkali paddy fields in North eastern China.
Qu et al.
The paper describes experiments of soil materials closely associated with plant roots compared to soil that is not closely associated with plant roots. The experiments looked at soil chemical properties, microbial populations and activity, and enzyme activity. The paper does identify several clear differences between soil materials closely associated with plant roots and soil materials not influenced by plant roots. However, the paper has a number of deficiencies that will need to be addressed before publication.
The paper does not provide a clear definition of what is understood by the term rhizosphere. This lack of a clear definition is exemplified by the experimental procedure which does not clearly demonstrate that the soil material upon which the experimental measurements is indeed “Rhizosphere material”. Fortunately, there are indications it is an approximation to the rhizosphere material for the rice plants. The authors need to reconcile existing definitions of rhizosphere in the scientific literature with their own use of the term and justify their experimental procedure to sample rhizosphere material from the soil.
The references below may assist
Husinger et al. (2005). Rhizosphere geometry and hetereogenity arising from non-mediated physical and chemical processes. New Phyologist.168, 293 = 303
Brimecombe et al, (2001). The effect of root exudates on rhizosphere microbial populations. In R Pinton et al. eds. The Rhizosphere. Marcell Dekker, New York.
Watt et al. (2006). Rhizosphere biology and crop productivity – a review. Australian Journal of Soil Research 44, 299 – 317.
The overall presentation of data needs some rethinking to sharpen it and make the results clearer. The results are there and it would only help the paper if the results are presented more visually and clearly.
The conclusion is flat and fails to present what seems to be some of the major results! The authors need to rethink about what the results of the paper really showed and what the key points were.
Specific comments
Line 27
Should indicate if the results showed a decrease or increase, not just a significant effect.
Line 37
Doubt if there are only three major areas of soda, saline alkali soils in the world?
Line 45
More encompassing and demonstrative definition of rhizosphere is required. Why is the rhizosphere different? How does it differ in soil chemistry and biology?
Line 75
Should provide a soil classification based on the World Reference Base.
It states that P1, P2 and P3 are different. Explain more about why they differ? It appears P3 is more alkaline? Does it occupy a different part of the landscape?
Table 1
Essential in this case to add some data on the EC, pH and ESP of the basic soil types at P1, P2 and P3. There is no indication of the depths of the samples taken. Without a depth indicated, the data is almost useless.
Line 86
How were the sampling sites chosen? Randomly picked? If a shovel sample was taken, to what depth, a standard depth? The assumption seems to have been made that soil close to or attached to the root was “rhizosphere materials”. How valid is this. Some authors suggest the rhizosphere can be as thin as 0.3 mm for some plants and soils or up to 1.0 mm for other soils and plants. Further explanation and justification for sampling procedure is necessary. Presumably soil material clinging to roots was assumed to rhizosphere material?
Lines 98 – 102
Messy presentation. Suggest maybe:
“…..pH in water in a 1:5 suspension and Exchangeable Sodium Percentage (ESP). the ESP was calculated as:
ESP = 100* exchangeable N+ [cmol(+)/kg]/cation exchange capacity [cmol(+)/kg].”
Lines 106 to 109
Clarify how the distinction was made between soil organic carbon and soil inorganic carbon. The analyser presumably converts all carbon to CO2 regardless of whether it is organic or inorganic. More explanation needed.
Line 150
No need for new paragraph.
Line 167
Suggest. “ ………rhizosphere soil were lower than…….”
Table 2
Difficult to read and assess. Suggest putting rhizosphere and non-rhizosphere together for each of the plots P1, P2 and P3. Need to put some thought into presenting results, but the rhizosphere v non-rhizosphere is the most important comparison.
Note the standard units for EC are dS/m which are the same numerical value as mS/m.
These values of EC1:5 are only moderately saline and do not support the proposal that these are highly saline soils. However, it is critical that the depth from which these samples were taken be identified. No depth is mentioned in this table. If these are EC1:5 for surface soils, then salinity would potentially be of concern for the soil profiles. Essential to identify the depths of the samples.
Figure 1
None of the 4 graphs have been identified as a, b, c and d? Add these.
Figure 2
Suggest different presentation to enable comparison of rhizosphere v non-rhizosphere soil material. Produce three graphs, one each for P1, P2 and P3 rather than current presentation. This would emphasise the results showing rhizosphere effects.
Table 5 is difficult to follow.
Instead of a tiny superscript to identify the major carbon substrates, use a capital letter symbol in front of each substrate. E.g. AA:L-threonine and CH:Pyruvic acid.
Figure 3 is poor and convert to standard histogram.
Line 253 – 254
The RDA analysis is complex and difficult to interpret. Needs a better explanation. One feels that the RDA analysis has too many variables and processes being assessed. It attempts to analyse too many things at once.
Why is there no comment that all the Rhizosphere sites appear to be close together, while the non-rhizosphere sites are wide apart in the RDA analysis. Some comment or explanation needed.
Lines 273 – 277
All redundant and unnecessary?
Line 323
Saline soil does not necessarily have a high pH, only if carbonates present. If only chlorides, can be more neutral.
Conclusion
Seems to emphasise some of the secondary outcomes from the experiments? Didn’t the rhizosphere have significantly different soil chemical properties to the non-rhizosphere material. Reconsider what are the conclusions from the study that are the most important?
Author Response

(The authors gave the same response as above.)

Reviewer 3 Report
The work needs to be improved by clarifying that
the microflora data refer to the cultivable fraction.
Therefore it is necessary to write this in the title and in the text,
as indicated in the attached pdf.

Author Response

(The authors gave the same response as above.)

Reviewer 4 Report
The manuscript titled “ The Influence of Rhizosphere and Non-rhizosphere Soil Physico-Chemical Properties on Enzyme Activities and Microorganism in Saline–Alkali Paddy Fields of Northeast China“. I find the idea interesting and in line with the aim of the journal. I have some concerns about the experimental set-up to justify what the authors claim. Moreover, the rationale behind some of the data presented was not entirely clear. I also recommend to the authors to improve their references by conducting a more extensive review on international literature. Particularly, in the introduction statements are not supported by the references selected by the authors. The logic of some sentences is also questionable. Below is my point to point analysis of the manuscript.
- I suggest to modify the title should be more crisp and brief.
- The introduction is very short it should reflect the proper background of the study, which I find is missing.
My main concern of a manuscript is the statical test.
what is the value of n while calculating ANOVA? Author mention n= 3
n Value (3) used in the manuscript is too few to examine normal distribution of variables in the sample, however, Shapiro-Wilk test is appropriate for samples from 3 to 5000 but for the lesser value of n, it receives the non-normal distribution. Thus ANOVA that is parametrical test is incorrect for such small samples.
The author should mention the data set that does not pass the normality test.
What method was used in exponential transformation, if it was done with the help of software mention the name of the software?
Secondly, the error bar in the figure correspond to the SD which does not make any sense, that is merely for decorative purpose. I highly recommend using confidence interval instead of Standard deviation, in the error bar.
- letters indicating a significant difference in figure no. 1 are calculated by SPSS or Origin.
- From the statement, it seems that letters are calculated using SPSS so in that case letters are manually placed over the graph?
I have doubt with the stat done as the statistic is the backbone of any research study.
- Highlighting the data set that donot follow the normal distribution
- Explaining how ANOVA was calculated for non-normal distribution.
Although the study is interesting and could be useful for a certain group of the scientific community, therefore, I would suggest improving the manuscript, giving a chance for the next round, because the subject is interesting
Author Response

(The authors gave the same response as above.)

Round 2
Reviewer 4 Report
The author has done a great job in modifying the manuscript. I am fully satisfied with the author reply. The manuscript does not need further revision. I recommend to accept it in the current form.
I congratulate the author for a nice piece of work.